



# $^{129}$Xe Ultrafast Z-spectroscopy enables micromolar detection of biosensors on a 1T benchtop spectrometer

Kévin Chighine[1], Estelle Léonce[1], Céline Boutin[1], Hervé Desvaux[1], and Patrick Berthault[1]

[1]Université Paris-Saclay, CNRS, CEA, Nanosciences et Innovation pour les Matériaux, la Biomédecine et l'Energie, 91191, Gif-sur-Yvette, France

**Correspondence:** Patrick Berthault (patrick.berthault@cea.fr)

**Abstract.** The availability of a benchtop NMR spectrometer, of low cost and easily transportable, can allow detection of low quantities of biosensors, provided that hyperpolarized species are used. Here we show that the micromolar threshold can easily be reached, by employing laser-polarized xenon and cage-molecules reversibly hosting it. Indirect detection of caged xenon is made *via* chemical exchange, using ultrafast Z-spectroscopy based on spatio-temporal encoding. On this non-dedicated low-field spectrometer, several ideas are proposed to improve the signal.

## 1 Introduction

In this sad period overshadowed by pandemia, among the analytical methods aiming at imaging the lung-blood transfer, hyperpolarized xenon NMR/MRI increasingly interests the '*in vivo*' scientific community. While xenon nuclear polarization can easily be boosted *via* spin-exchange optical pumping (Walker and Happer, 1997, SEOP), the other interest of this noble gas for NMR is that it exhibits a wide chemical shift range (more than 320 ppm for the monoatomic species) and is soluble in most biological fluids. Therefore, xenon is a powerful exogenous probe of the functioning of the air-blood barrier (Driehuys et al., 2006). Moreover, it is prone to open the way to molecular magnetic resonance imaging. In an approach pioneered by A. Pines and co-workers (Spence et al., 2001), xenon is reversibly encapsulated in molecular systems that are functionalized with biological ligands. This two-step procedure, where the bioprobe is first introduced and then hyperpolarized xenon is delivered, benefits from the difference in resonance frequency between bound xenon and free xenon (in the gas phase or in the dissolved phase, cf. Berthault et al. (2009)).

In this method, most of the studies used cryptophane derivatives as xenon hosts, as despite a complex synthesis they are functionalizable by ligands (Brotin and Dutasta, 2009). For instance, to our knowledge, only one example of chemical functionalization of a xenon host other then cryptophane - a cucurbituril - has been reported in the literature (Truxal et al., 2019). The approach of $^{129}$Xe NMR-based biosensing using functionalized cryptophanes has yet been successfully applied *in vitro* for detection of small analytes (Tassali et al., 2014; Dubost et al., 2014; Jeong et al., 2015; Yang et al., 2016; Guo et al., 2016; Yang et al., 2017), of large biosystems (Wei et al., 2006; Chambers et al., 2009; Boutin et al., 2011; Rose et al., 2014; Taratula et al., 2015; Khan et al., 2015; Riggle et al., 2017; Milanole et al., 2017; Schnurr et al., 2020), or of change of physiological conditions: temperature (Schröder et al., 2008; Schilling et al., 2010) or pH (Léonce et al., 2018). To date, it has however never



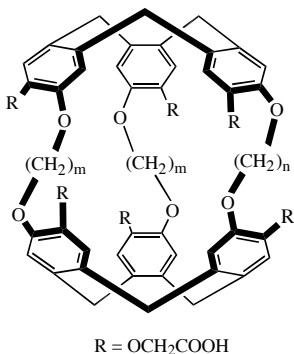

R = OCH$_2$COOH

**Figure 1.** Generic structure of the cryptophanes used in this study. For compound **1**: m = 2 and n = 3; for **2**: m = 3 and n = 2.

been used *in vivo*; only a proof of concept has been performed on rats using a non-functionalized cucurbituril (Hane et al., 2017). Several difficulties or obstacles have delayed *in vivo* applications, among which obviously the lack of sensitivity.

We made the remark that in a pre-clinical environment, it could be very useful to test the behaviour of such bioprobes in NMR, using a benchtop spectrometer, less cumbersome and less expensive than a high-field spectrometer. It could be placed very close to the optical pumping setup, working in flow or batch modes. The present work aims at assessing the feasibility of

the detection of low concentrations of $^{129}$Xe NMR-based biosensors using a non-dedicated benchtop spectrometer. After a brief description of our spin-exchange optical pumping setup working in the batch mode, direct and indirect detection techniques are studied both at low and high magnetic field. Theoretical considerations are given and practical ways of improvement are analyzed.

## 2    Results

## 2.1    Molecular systems

For such a study, we decided to use two water-soluble cryptophanes, synthesized by the group of T. Brotin at ENS Lyon, and previously characterized (Huber et al., 2006). Their generic structure is depicted in Fig. 1. In cryptophane **1** the two cyclotriveratrilene bowls are connected by two O-(CH$_2$)$_2$-O linkers and a O-(CH$_2$)$_3$-O linker; in cryptophane **2** they are connected by two O-(CH$_2$)$_3$-O linkers and a O-(CH$_2$)$_2$-O linker. Xenon inside these cage-molecules resonates at 52 ppm (for Xe@**1**) and 42

40    ppm (for Xe@**2**) if one calibrates the signal of xenon free in water at 196 ppm. These two molecular systems could be used as probes for pH, as the chemical shift of caged xenon varies as a function of the concentration in H$^+$ ions (Léonce et al., 2018).



## 2.2 Addition of hyperpolarized xenon into the samples

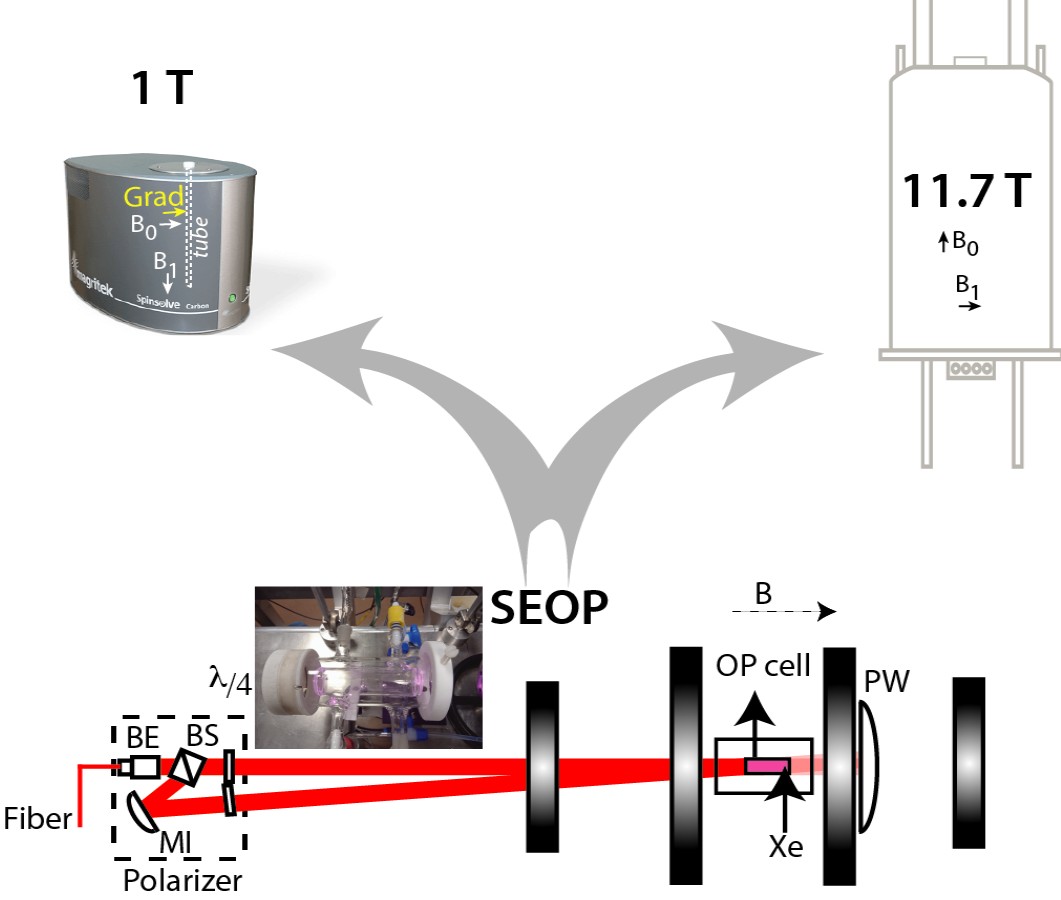

**Figure 2.** Principle of the experiments. Laser-polarized xenon is produced via Spin-Exchange Optical Pumping in the batch mode with the setup described in Chauvin et al. (2016). BE: Beam Expander; BS: polarization Beam Splitter cube; MI: mirror; $\lambda/4$: quarterwave plates; PW: Powermeter. The black vertical rectangle represent the coils, which deliver a magnetic field $B$ colinear to the photon beam. A picture of the optical pumping cell is given in inset. After some minutes of optical pumping, frozen hyperpolarized xenon is transported to the NMR spectrometers. The disposition of the NMR tube inside the magnet, the static and radiofrequency fields as well as the axis of the gradient are indicated on the benchtop spectrometer.

Xenon enriched at 83 % in isotope 129 was polarized *via* spin-exchange optical pumping using a home-built setup already described in Chauvin et al. (2016). After about 10 minutes optical pumping, hyperpolarized xenon was collected frozen and

transported immersed in liquid nitrogen inside a 0.3 T solenoid feeded by a car battery. Then, in the fringe field of the unshielded 11.7 T magnet, xenon was heated and transferred to the capped NMR tube containing the sample, thanks to a vacuum line and a hollow spinner filled with liquid nitrogen. This procedure enabled us to transfer all xenon above the solution without freezing





it. Finally fast heating and vigorous shaking of the tube speeded up the dissolution of the noble gas into the sample of interest. This method, proved efficient for high magnetic field experiments, is not directly translatable to the NMR experiments in the benchtop spectrometer. All attempt to use the fringe field of the permanent magnet during the xenon transfer and shaking step led to deceiving results in terms of remaining polarization. This was expected as the magnet, of the Halbach type, delivers an horizontal static magnetic field, and thus xenon crosses areas of null field during its transfer. The least bad solution was thus to carry out the introduction of polarized xenon inside the NMR tube in the fringe field of the superconducting magnet, then to quickly shake the tube in this magnetic environment before transporting it to the benchtop spectrometer. The following shaking, carried out as close as possible to the permanent magnet, leads to a faster depolarization.

Without optimization other than the setting of the quarterwave plates, this robust setup gave us a useable polarization of *ca.* 0.15, as measured in the gas phase on the 11.7 T spectrometer by comparison with the thermal equilibrium NMR signal.

The pressure in the NMR tube on top of the solution was ca. 1 bar, as estimated post NMR experiment by weighing the tube before and after degassing.

## 2.3 Direct detection of $^{129}$Xe NMR-based biosensors

Figure 3 displays a comparison of one-scan $^{129}$Xe NMR spectra of the same sample of cryptophanes dissolved at a concentration of 77 $\mu$M in D$_2$O, at 11.7 and 1 Tesla (same NMR tube). The two spectra were recorded with different xenon batches, but acquired in the same experimental conditions.

Not surprisingly, the signal-to-noise ratio is better at 11.7 Tesla than at 1 T. Obviously if the xenon nuclear polarization should be similar for the two experiments as it directly derives from the optical pumping step, the signal-to-noise ratios are expected to be very different due to the detection part. Indeed, in classical NMR, it is the magnetic induction, proportional to the temporal derivative of the component of the magnetization perpendicular to the static magnetic field that is detected, and thus the Larmor frequency on the one side, and all geometrical and electronic parameters defining the sensitivity (coil geometry, useful volume, quality factor, tuning frequency, filter and digitization...) on the other side would have to be considered for comparing the signal-to-noise ratio.

From Fig. 3, several remarks can be made. While at high field the $^{129}$Xe NMR spectrum displays one signal at 196 ppm corresponding to xenon free in water and two distinct signals at high field corresponding to caged xenon (Xe@**1** at 52 ppm and Xe@**2** at 42 ppm according to Huber et al. (2006)), at low static magnetic field one of the latter signals disappears or is hardly observable. The full-width at half-maximum of the Xe@**1** signals were roughly measured to 44 Hz and 20 Hz at 11.7 and 1 Tesla, respectively. The presence of a narrow peak at 11.7 T and a very broad line at 1T for Xe@**2** reveals the weakness of the direct detection approach, even if one uses a fast repetition 'frequency-selective pulse - detection' (Berthault et al., 2008). When the lines are so broad and the signal-to-noise so low, each of these acquisitions brings a lot of noise and the resulting signal is difficult to distinguish from the baseline.

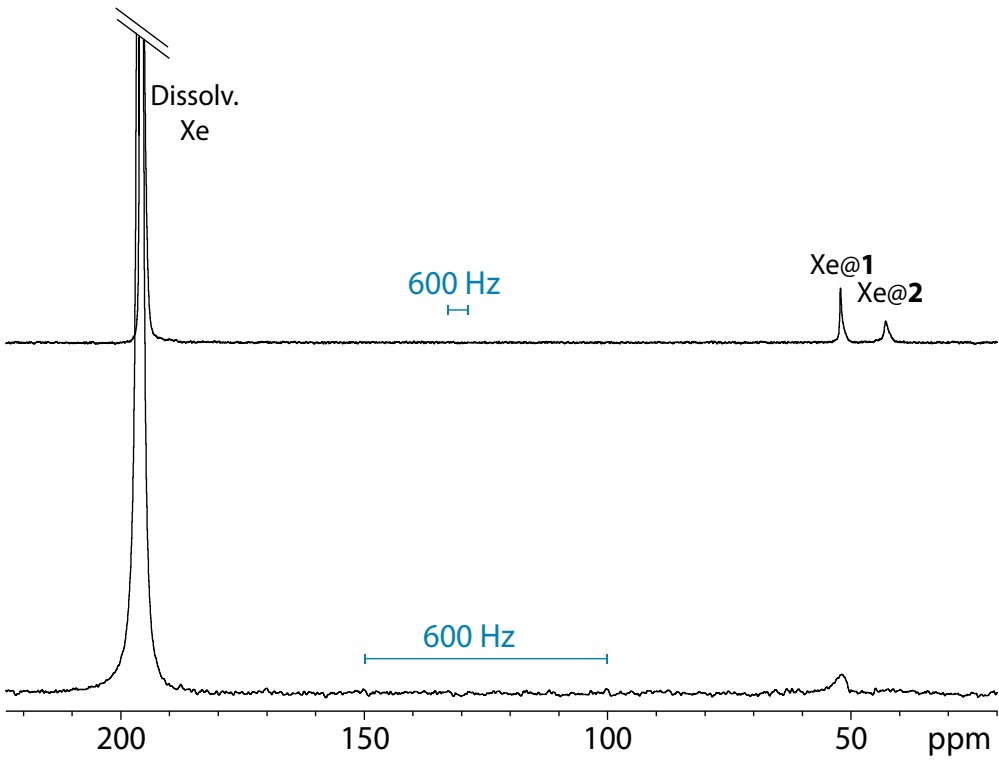

**Figure 3.** One-scan $^{129}$Xe NMR spectra of the noble gas into a water solution of cryptophanes **1** and **2**, both at 77 $\mu$M and 295 K, with the same pulse flip angle. Top spectrum performed at 11.7 Tesla ; bottom spectrum performed at 1 Tesla.

## 2.4 Indirect detection

It is well known that the detection of diluted species in exchange with a main spin reservoir can be facilitated by taking advantage of this exchange. This has given rise to the Chemical Exchange Saturation Transfer (CEST) experiments, which principle lies in the rf saturation at the frequency of these diluted species and the observation of the subsequent loss of magnetization of the main reservoir due to the exchange (see for instance Vinogradov et al. (2013)). Schröder and co-workers have extended this approach to hyperpolarized species, creating the Hyper-CEST sequence (Schroder et al., 2006).

But sequences of the CEST family are usually reserved to high magnetic fields. For instance, for detection of metabolites such as glutamate or carnosine, $^1$H CEST is an efficient method (Cai et al., 2012; Bodet et al., 2015). But it would not come to mind to use a CEST sequence with a low magnetic field, as the frequency splitting between the two environments in exchange is only ca. 3.3 ppm (140 Hz at 1 Tesla). However, due to the very wide chemical shift range of xenon, this can be envisioned in the case of $^{129}$Xe NMR-based biosensors.





Due to the huge advantage of a spectrum-per-spectrum averaging instead of a point-per-point averaging for hyperpolarized species, we decided to turn to the ultra-fast version of the CEST experiment, initially proposed by Jerschow and co-workers in [1]H (Xu et al., 2013). In this sequence, saturation is applied in the presence of a gradient, which amounts to saturate only a slice of the sample. After a read pulse, the receiver is open in the presence of a second gradient which decodes the profile of the sample along it. By subtracting two experiments recorded with and without saturation, one obtains the Z-spectrum. Note

that this sequence can be combined to localized spectroscopy or imaging schemes, see for instance Döpfert et al. (2014b); Xu et al. (2016); Liu et al. (2016).

In the past, the Ultra-Fast Z-spectroscopy (UFZ) experiment was successfully applied in hyperpolarized [129]Xe NMR for the study of biosensors by us and others (Boutin et al., 2013; Döpfert et al., 2014a), and also for detection of low amounts of biological cells (Berthault and Boutin, 2015)). For this study, we used the pulse sequence depicted in Fig. 4a. It contains an

rf offet switch: the first value, $\omega_B$, is centered on the Xe@cryptophanes region (*i.e.* around 45 ppm), while the second value, $\omega_A$, is applied on-resonance with the free xenon frequency. Such a sequence is not intended to provide the full Z-spectrum (saturation at the free xenon frequency would be disastrous for hyperpolarization), but is prone to reveal the presence of caged xenon (cf. Boutin et al. (2013) for details).

Figure 4b displays an [129]Xe UFZ-spectrum recorded on the cryptophane mixture at 11.7 Tesla and 295 K.

The apparent chemical shift splitting between the two dips can be measured at 25.4 ppm, which well corresponds to the real chemical shift splitting of 10 ppm when the ratio between the saturation and acquisition gradients is considered (see Theorerical part). The dip corresponding to Xe@**2** is broader than that of Xe@**1**, which can be explained by a faster xenon in-out exchange.

Figure 5 displays an [129]Xe UFZ-spectrum recorded on the cryptophane mixture at 1 Tesla and 295 K. For the spatial encoding of the [129]Xe UFZ-spectroscopy experiment, on the Magritek SpinSolve C spectrometer we had the choice to use

the shim system to create pulsed field gradients in the $x$, $y$ or $z$ directions, or to use the installed gradient system. The latter gradient is oriented horizontally while the main axis of the NMR tube is vertical (see Fig. 2). It could have been interesting to use the shim along the NMR tube axis, as the magnetization profile would have been along the largest dimension of the tube and would have given a flat profile. However, for stability reasons we decided to use the nominal gradient system. This explains the rounded shape of the magnetization profile envelope, corresponding to an axial projection of the tube.

The gradient strength was firstly calibrated by acquiring a 1D axial profile of the tube in [1]H. Then we chose the value of the gradient simultaneous to [129]Xe saturation, $G_{sat}$, keeping in mind that it had to induce a spectral expansion of the signal saturation smaller than the difference between the free xenon frequency $\omega_A$ and the caged xenon frequency $\omega_B$, thus filling the condition:

$$\gamma.G_{sat}.d < |\omega_A - \omega_B|$$

with $d$ the inner diameter of the tube. In our case, $d = 0.43$ cm, $\gamma = 1178.8$ s$^{-1}$G$^{-1}$, $|\omega_A - \omega_B| \simeq 1800$ Hz induces that $G_{sat}$ must be lower than 3.55 G.cm$^{-1}$.

The profile of the UFZ-spectrum reveals two dips corresponding to xenon in the two cryptophanes, with an apparent frequency splitting of about 40 ppm, which is the expected value for Xe@**1** and Xe@**2** given the ratio $G_{sat}/G_{acq}$. The rf carrier





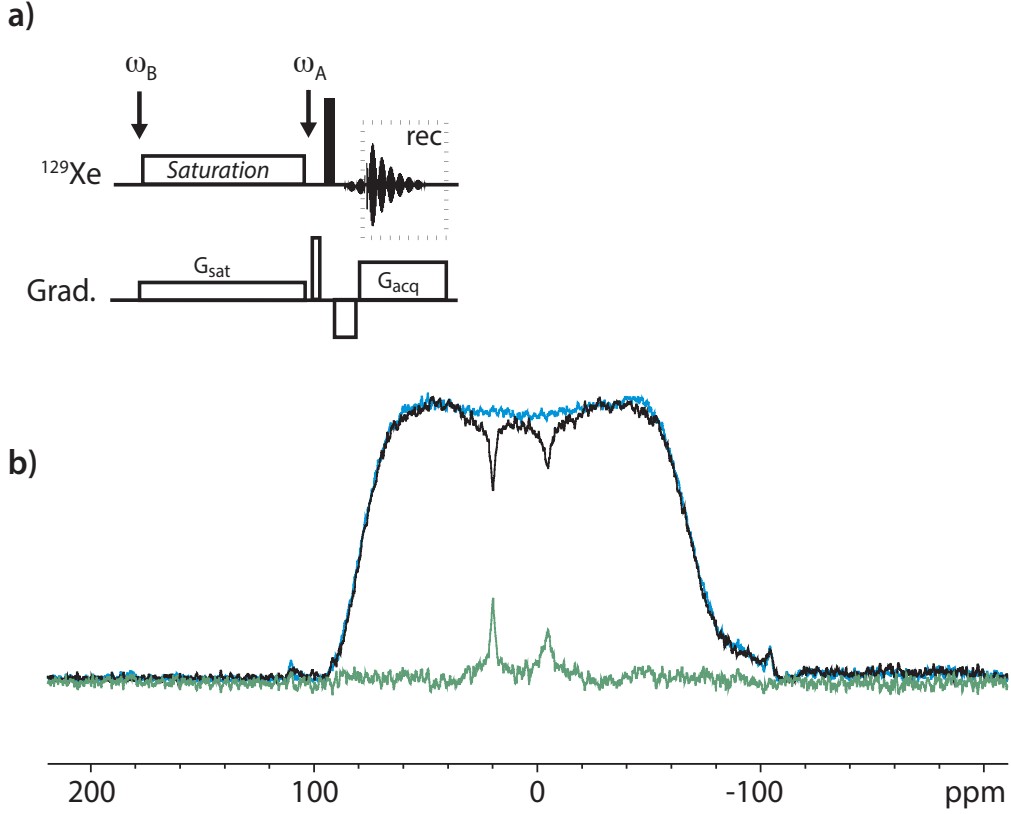

**Figure 4.** a) Pulse sequence used for the $^{129}$Xe UFZ-spectroscopy. $\omega_B$ and $\omega_A$ denote the rf offset placed in the region of caged xenon and at the frequency of free xenon, respectively. b) $^{129}$Xe UFZ-spectrum of the noble gas into the solution containing the cryptophane mixture at 11.7 T and 295 K. In black: the profile obtained after 2 s saturation at a field strength $B_1 = \omega_1/\gamma = 7\mu$T (*On* experiment); in blue: same profile without saturation (*Off* experiment); in green: *Off - On*. Other important parameters: $G_{\text{sat}}$= 3.5 G.cm$^{-1}$; $G_{\text{acq}}$= 9 G.cm$^{-1}$.

having been placed on resonance with the Xe@**1** frequency, we have assigned the dips as displayed on Fig. 5. The apparent

reverse frequency axis is due to the relative sign of the two gradients. Obviously the separation between the dips is less net than at high field, and to confirm our assignment we have performed the same experiment with another xenon batch, simply inverting the sign of $G_{\text{sat}}$. The corresponding two-scan UFZ-spectrum is displayed in Fig. 6.

In order to model the behavior of the Z-spectra as a function of the experimental parameters, we have simulated the Hyper-CEST experiment (see Theory Section). As input of these simulations, $R_{1_A}$ can be measured in a experiment consisting in

a series {small flip angle pulse - detection}, by taking into account the flip angle. $R_{2_A}$ can be roughly estimated from the linewidth of $A$, but with a large uncertainty. $f$ can be directly determined from a simple $^{129}$Xe spectrum as it is the ratio of the signal $B$ to the signal $A$. But the parameter which remains difficult to predict or estimate is $k_{\text{out}}$.

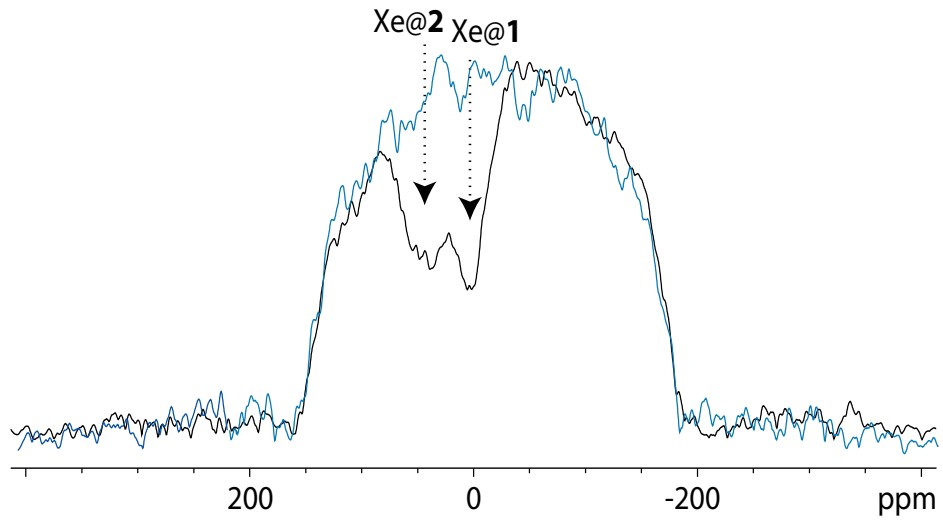

**Figure 5.** $^{129}$Xe UFZ-spectrum of the noble gas into the solution containing the cryptophane mixture at 1 T and 295 K, using the same pulse sequence as for Fig. 4 (sum of two experiments with saturation of 2 s and 4 s). The saturation offset was placed at the Xe@**1** frequency. In black: the profile obtained after saturation at a field strength $B_1 = \omega_1/\gamma = 0.87$ $\mu$T (*On* experiment); in blue: same profile without saturation (*Off* experiment); Other important parameters: $G_{\text{sat}}$= 2.1 G.cm$^{-1}$; $G_{\text{acq}}$= 8.4 G.cm$^{-1}$.

In a first step, we used Equations 13 and 11 to simulate the trends on the $^{129}$Xe UFZ spectra recorded at 11.7 and 1 T (Fig. 7). In Fig. 7a, in order to evidence the effect of the Larmor frequency, all other parameters have been kept identical ($f$ is the fraction of caged xenon; $k_{\text{out}}$ is the xenon out rate):

$f = 0.04$ ; $k_{\text{out}} = 50$ s$^{-1}$ ; $\omega_A = 196$ ppm ; $\omega_B = 52$ ppm ; $R_{1_A} = 0.01$ s$^{-1}$; $R_{2_A} = 50$ s$^{-1}$ ; $\omega_1 = 82.5$ Hz (7 $\mu$T) ; $t_{\text{sat}} = 1$s.

This simulation shows that the rf saturation strength must be reduced on the low field spectrometer in order to keep a flat baseline, due to the low difference between the free and bound xenon resonance frequencies.

Figure 7b displays the simulation for more realistic experimental conditions. The parameters were the following :

- Parameters common to both fields: $f = 0.04$ ; $k_{\text{out}} = 50$ s$^{-1}$ ; $\omega_A = 196$ ppm ; $\omega_B = 52$ ppm

- Spinsolve (Larmor frequency = 12.09 MHz): $R_{1_A} = 0.01$ s$^{-1}$; $R_{2_A} = 20$ s$^{-1}$ ; $\omega_1 = 10$ Hz (0.87 $\mu$T) ; $t_{\text{sat}} = 3$ s

- Avance (Larmor frequency = 138.36 MHz): $R_{1_A} = 0.01$ s$^{-1}$; $R_{2_A} = 50$ s$^{-1}$ ; $\omega_1 = 82.5$ Hz (7 $\mu$T) ; $t_{\text{sat}} = 1$ s





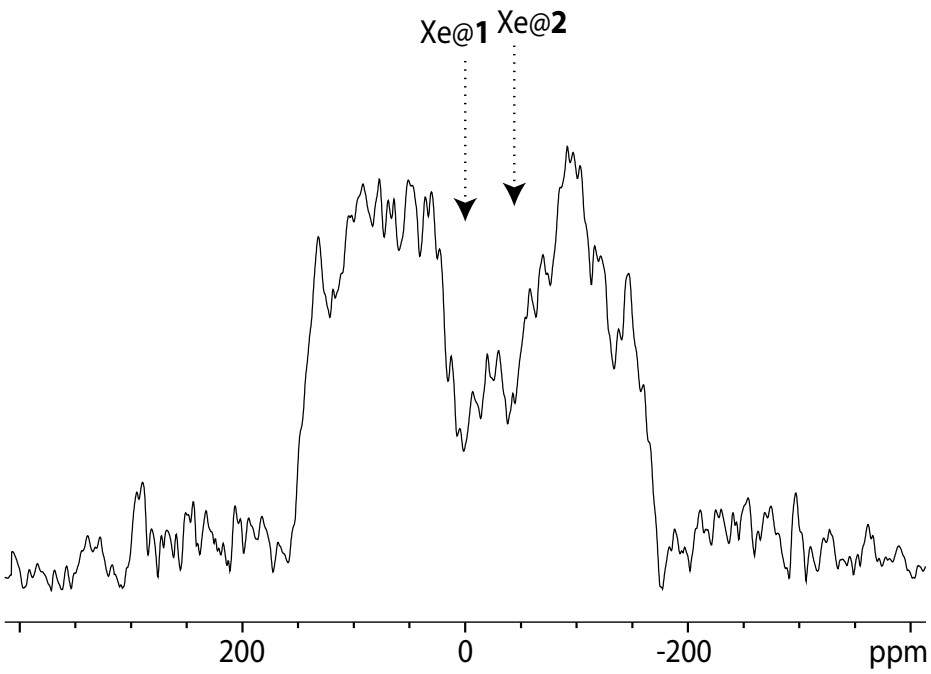

**Figure 6.** $^{129}$Xe UFZ-spectrum of the noble gas into the solution containing the cryptophane mixture at 1 T and 295 K, using the same pulse sequence as for Fig. 5, except that the saturation gradient is opposite: $G_{\mathrm{sat}}$= -2.1 G.cm$^{-1}$; $G_{\mathrm{acq}}$= 8.4 G.cm$^{-1}$. Saturation delay: 3 s.

Comparison between the two $^{129}$Xe Z-spectra simulated with realistic conditions evidences firstly the effect of the smaller frequency splitting on the Magritek. Despite lower relaxation rates and despite a weaker saturation strength at low field, the

direct relaxation term $\lambda_{\mathrm{direct}}(\omega_i, \omega_1)$ has a larger influence on the baseline in the Xe@cryptophane region (purple curve). This term decreases the signal by about three percent. This reveals that the saturation field strength must stay small to ensure the success of the experiment. Secondly, such a small $\omega_1$ requires a longer saturation time. With a saturation time $t_{\mathrm{sat}}$ three times longer than the one used at high field, the dip in the Xe@cryptophane region has almost the same depth.

  The second simulation considered two cryptophanes with characteristics close to the ones used in the present study. The

parameters of the $^{129}$Xe Z-spectra were those of the real experiment on the benchtop spectrometer:

- Parameters common to both cryptophanes: Larmor frequency = 12.09 MHz ; $R_{1_A} = 0.1$ s$^{-1}$; $R_{2_A} = 10$ s$^{-1}$ ; $t_{\mathrm{sat}} = 3$ s

- Cryptophane 1: $f = 0.037$ ; $k_{\mathrm{out}} = 50$ s$^{-1}$; $\omega_B = 52$ ppm

- Cryptophane 2: $f = 0.025$ ; $k_{\mathrm{out}} = 100$ s$^{-1}$; $\omega_B = 42$ ppm



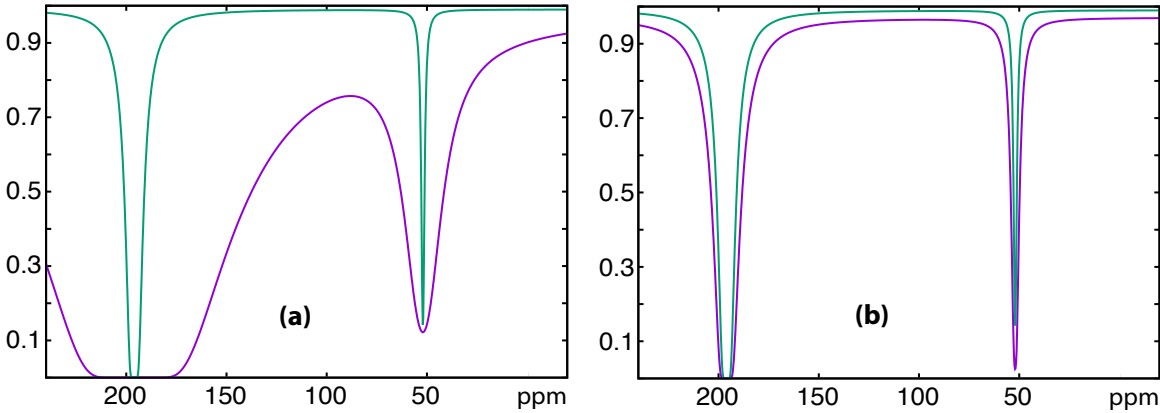

**Figure 7.** Simulation of $^{129}$Xe Z-spectra at 11.7 (green) and 1 (purple) Tesla. (a): Considering same relaxation times, same rf field strength and duration of the saturation ; (b): Considering a saturation of lower amplitude but larger duration for the low-field experiment. Details of the conditions for the simulation: see text.

Figure 8 displays the effect of the saturation strength $\omega_1$ on the aspect of the $^{129}$Xe Z-spectrum of the cryptophane mixture

(only the 0 - 100 ppm region is displayed). At low saturation strength, the signal of the cryptophane with the highest $f/k_{\mathrm{out}}$ is predominant, as expected (see theoretical part). Then, when $\omega_1$ increases the second dip becomes more pronounced, larger than the first one due to the higher $k_{out}$ value (Equation 9). But increasing further $\omega_1$ leads to significant lowering of the baseline, thus of the contrast, and the signals are less separated. Thus a compromise has to be found to favor detection of the dips and maximize their separation. With the current conditions of the simulation, a value of $\omega_1$ between 10 and 20 s$^{-1}$ seems the best.

These simulations have helped us to find the best experimental conditions.

In such an approach, the detection threshold can further be lowered, maybe at the price of a lack of discrimination between two minor sites in exchange with the main signal. As an example that does not seek to be a record, this experiment has been repeated with a solution containing cryptophane **2** at a concentration of 19.2 $\mu$M. Figure 9 displays the $^{129}$Xe UFZ profile.



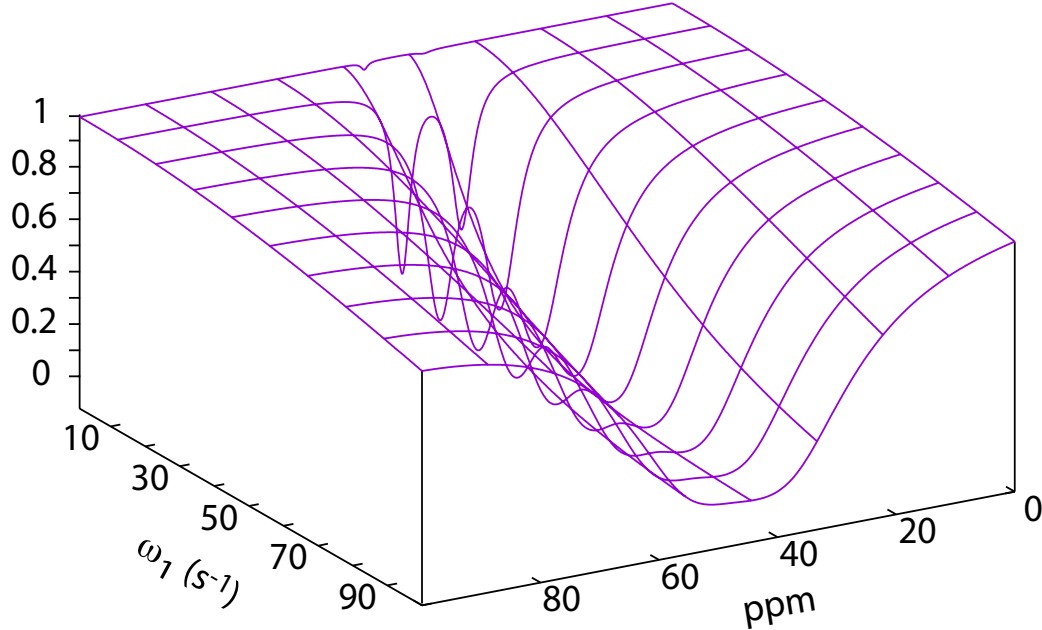

**Figure 8.** Simulation of $^{129}$Xe Z-spectra at 1 Tesla for a mixture of two cryptophanes according to the saturation field strength. Conditions for the simulation: see text.

With respect to the previous experiments, only the saturation field strength has been slightly increased from 0.87 to 1.24 $\mu$T
($\omega_1$ = 10.3 to 14.7 s$^{-1}$). The dip corresponding to caged xenon appears clearly.

## 3 Materials and Methods

### 3.1 Preparation of the samples

A mother solution of cryptophane **1** was prepared by dissolution of 0.64 mg of the powder in 500 $\mu$L D$_2$O and 20 $\mu$L NaOD
0.1 M. The same preparation was done for cryptophane **2**. 50 $\mu$L of each of these solutions were mixed and then the solution
was diluted by a factor 5 in D$_2$O, giving a resulting concentration of 77 $\mu$M for cryptophane **1** and 78 $\mu$M for cryptophane
**2**. The solution was placed in a 5 mm-NMR tube equipped with a valve. The tube was degassed before each inyroduction of
hyperpolarized xenon by three cycles of helium bubbling then evacuation.

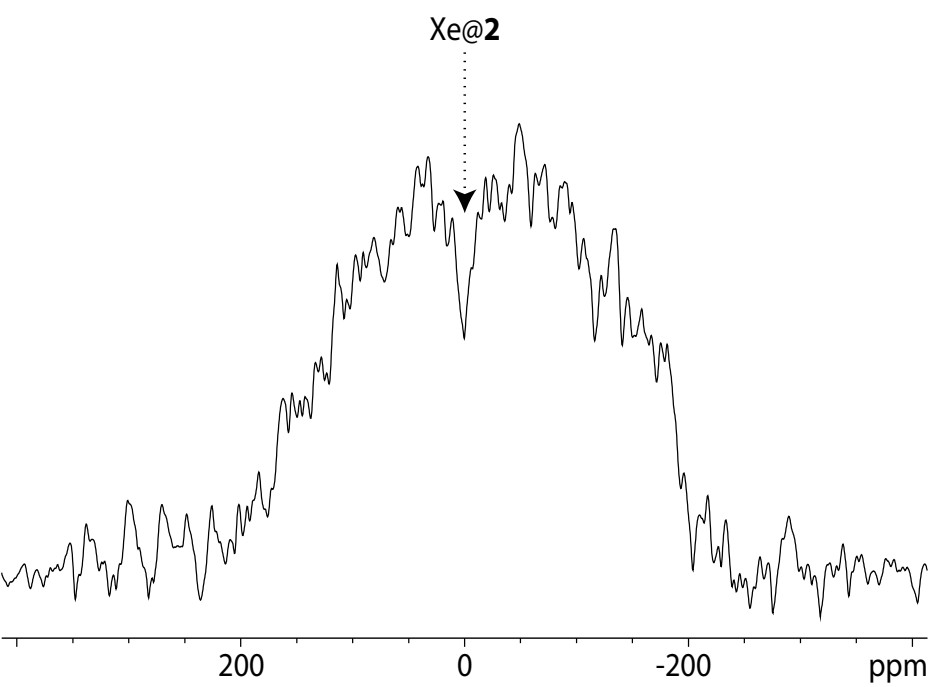

**Figure 9.** $^{129}$Xe UFZ-spectrum of the noble gas into the solution containing cryptophane **2** at 19.2 $\mu$M. Magnetic field: 1 T ; temperature 295 K, same pulse sequence as for Fig. 4 (one scan). $\omega_B$ was placed at the Xe@**2** frequency. Profile obtained after 3 s saturation at a field strength $B_1 = \omega_1/\gamma = 1.24$ $\mu$T. $G_{\text{sat}}= 2.1$ G.cm$^{-1}$; $G_{\text{acq}}= 8.4$ G.cm$^{-1}$.

## 3.2 Observation of $^{129}$Xe on the 11.7 T spectrometer

The experiments were performed at 295 K on a Bruker Avance II spectrometer equipped with a dual $^{129}$Xe-$^1$H probehead and
a GREAT3/10 three-axis gradient amplifier.

## 3.3 Observation of $^{129}$Xe on the benchtop spectrometer

The experiments were performed at 295 K on a Magritek SpinSolve Carbon spectrometer at a magnetic field of 1.02 T. The $X$ channel of the low-field spectrometer being initially optimized for $^{13}$C (*i. e.* 10.9 MHz), it was necessary to lower the temperature of the permanent magnet in order to approach the $^{129}$Xe Larmor frequency (12.1 MHz). Obviously, the situation is
not optimal for $^{129}$Xe, and the value of the 90° pulse was measured to 150 $\mu$s at full power. This was calibrated with a sample of thermal xenon inside dodecane dopped with Gd$^{3+}$ ions. Note that at such low field, the paramagnetic doping was not very





efficient, and a $T_1$ of *ca.* 10 s was found for $^{129}$Xe in this solution. The same procedure was used to calibrate the $B_1$ strength value for saturation.

The SpinSolve software in expert mode was used to create the pulse programs that were not in the Magritek library. In particular the UltraFast Z-spectroscopy sequence was written and tested, first in $^1$H NMR, then in $^{23}$Na NMR, before the $^{129}$Xe NMR experiments.

In order to apply exactly the same processing to the data acquired at 1 and 11.7 T and to compare them safely, a program was written in Python to convert the FID recorded under Spinsolve into JCAMP-DX data readable by the Bruker software, Topspin.

## 4 Theory of the Hyper-CEST experiment

Let us consider a cryptophane solubilized in water at a sub-millimolar concentration. A small part of hyperpolarized xenon, when introduced in solution, will be reversibly caged in cryptophane. The $^{129}$Xe spectrum will thus exhibit two signals: free dissolved xenon (pool $A$ of magnetization $M_A^0$, giving rise to the main signal at $\omega_A$) and xenon inside the cryptophane (pool $B$ of magnetization $M_B^0$, very minor signal at $\omega_B$). Two types of xenon in-out exchange coexist: Simple dissociation:

$$195 \quad X + C \underset{k_-}{\overset{k_+}{\rightleftharpoons}} CX \qquad (1)$$

($X$ for xenon, $C$ for cryptophane), and degenerate (or kick-out) exchange, dependent on the ratio xenon concentration to cryptophane concentration (Korchak et al., 2016) :

$$CX + X^* \underset{k}{\rightleftharpoons} CX^* + X \qquad (2)$$

where the asterisk denotes the hyperpolarized state. As the two xenon pools have different resonance frequencies, these pro-
200 cesses can be seen as a unique exchange between free and encapsulated xenon, characterized by the rates $k_{in}$ and $k_{out}$ at steady-state:

$$k_{in} = k_{out} \frac{M_B^0}{M_A^0} = k_{out} \times f \qquad (3)$$

recalling that $f$ is the fraction of caged xenon.

The Hyper-CEST experiment consists in saturating with a (CW) rf irradiation of strength $\omega_1$ in the Xe@cryptophane region
(pool $B$), and detecting the influence on the main Xe signal (pool A). This has two consequences:

◇ A direct effect linked to the rf saturation at an offset $\Delta_i = \omega_i - \omega_A$ from the main resonance, which tilts the magnetization by an angle $\theta_i = \tan^{-1} \frac{\omega_1}{|\Delta_i|}$ with $\theta_i$ in $[-\pi/2 : \pi/2]$ (Desvaux and Berthault, 1999). The magnetization transverse to this effective field is averaged out by rf field inhomogeneity and transverse relaxation. The magnetization alogned with the effective field is going to relax. This effect, present even in the absence of exchange, is characterized by the depolarization



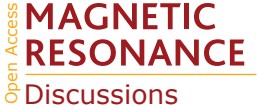

rate:

$$\lambda_{\text{direct}}(\omega_i, \omega_1) = R_{1_A} \cos^2 \theta_i + R_{2_A} \sin^2 \theta_i \tag{4}$$

where $R_{1_A}$ and $R_{2_A}$ are the longitudinal and transverse relaxation rates of free xenon, respectively. In the range of chemical exchange observed with cryptophanes $k = k_{\text{in}} + k_{\text{out}} \gg R_{1_A}, R_{2_A}$, given the low proportion of xenon caged in the cryptophane, one has:

$$R_1 = (1 - f)R_{1_A} + f R_{1_B} \simeq R_{1_A} \tag{5}$$

$$R_2 = (1 - f)R_{2_A} + f R_{2_B} \simeq R_{2_A} \tag{6}$$

◇ An indirect depolarization due to the saturation transfer from the pool $B$ to the pool $A$ (CEST effect). If the saturation is applied exactly on-resonance with the caged xenon frequency ($\omega_i = \omega_B$), with the assumption $k_{\text{out}} \gg R_2^B$ and $k_{\text{out}} \gg k_{\text{in}}$, the depolarization rate linked to saturation at the frequency of pool $B$ is given by (Zaiss et al., 2012; Kunth et al., 2014):

$$\lambda_{\text{on}}(\omega_1) \simeq f k_{\text{out}} \frac{\omega_1^2}{\omega_1^2 + k_{\text{out}}^2} \tag{7}$$

This has enabled Kunth and co-workers to distinguish three cases (Kunth et al., 2015):

- $\omega_1 \gg f k_{\text{out}} : \lambda_{\text{on}} \simeq f k_{\text{out}}$. *Maximum depolarization rate.*
- $\omega_1 = f k_{\text{out}} : \lambda_{\text{on}} \simeq f k_{\text{out}}/2$. *Fifty percent of the maximum possible depolarization rate.*
- $\omega_1 \ll f k_{\text{out}} : \lambda_{\text{on}} \simeq (f/k_{\text{out}})\omega_1^2$. *Parabolic behavior with the saturation strength.*

In order to understand the UFZ-spectroscopy experiment, one needs to consider the difference frequency between the frequency of the rf irradiation, $\omega_i$, and the frequency of pool $B$, $\omega_B$, which has a Lorentzian shape as a function of $\omega_i$ (Zaiss et al., 2012).

$$\lambda_{\text{CEST}}(\omega_i, \omega_1) = \frac{\lambda_{\text{on}}(\omega_1)\frac{\Gamma(\omega_1)^2}{4}}{\frac{\Gamma(\omega_1)^2}{4} + (\omega_i - \omega_B)^2} \tag{8}$$

with the full-width at half-maximum value of the depolarization rate given by:

$$\Gamma(\omega_1) \simeq 2\sqrt{\omega_1^2 + k_{\text{out}}^2} \tag{9}$$

Thus, the total depolarization raite is:

$$\lambda(\omega_i, \omega_1) = \lambda_{\text{direct}}(\omega_i, \omega_1) + \lambda_{\text{CEST}}(\omega_i, \omega_1) = R_{1_A} \cos^2 \theta_i + R_{2_A} \sin^2 \theta_i + \frac{\lambda_{\text{on}}(\omega_1)\frac{\Gamma(\omega_1)^2}{4}}{\frac{\Gamma(\omega_1)^2}{4} + (\omega_i - \omega_B)^2} \tag{10}$$

which can be re-written:

$$\lambda(\omega_i, \omega_1) = R_{1_A} \frac{1}{1 + \left(\frac{\omega_1}{\omega_i - \omega_A}\right)^2} + R_{2_A} \frac{\left(\frac{\omega_1}{\omega_i - \omega_A}\right)^2}{1 + \left(\frac{\omega_1}{\omega_i - \omega_A}\right)^2} + \frac{f k_{\text{out}}.\omega_1^2}{\omega_1^2 + k_{\text{out}}^2 + (\omega_i - \omega_B)^2} \tag{11}$$





If one considers the experiment at 11.7 Tesla with the usual modest saturation strength, $|\omega_i - \omega_A| \gg \omega_1$, Equation 11 can be simplified to:

$$\lambda(\omega_i, \omega_1) = R_{1_A} + \frac{f k_{\text{out}}.\omega_1^2}{\omega_1^2 + k_{\text{out}}^2 + (\omega_i - \omega_B)^2} \tag{12}$$

The z-magnetization after a saturation of duration $t_{\text{sat}}$ is :

$$Z(\omega_i, \omega_1) = M_A^0.\mathrm{e}^{-\lambda(\omega_i,\omega_1)t_{\text{sat}}} \tag{13}$$

**Specificity of the Ultra-Fast Z-spectroscopy**

The Ultra-Fast Z-spectroscopy (UFZ) consists in applying the rf saturation in the presence of a magnetic field gradient $G_{\text{sat}}$ and after a read pulse detecting the signal in the presence of another gradient $G_{\text{acq}}$. The obtained profile reflects the magnetization all along the sample in the $G_{\text{acq}}$ gradient direction with dips at positions where saturation is effective according to the rf offset and the gradient $G_{\text{sat}}$. Thus after subtraction from the profile acquired through the same sequence without saturation and application of a scaling factor of the intensity for taking into account relaxation at the transient high polarization, it corresponds to the Z-spectrum. After the read pulse, the acquisition in the presence of a gradient $G_{\text{acq}}$ makes that all spectral information is spread on a frequency range equal to $\gamma.G_{\text{acq}}.r$, where $r$ is the dimension of the sample along the gradient axis. In summary, two signals separated on a normal NMR spectrum by a value $\Delta\omega_{B_{12}}$ are now separated in the UFZ-spectrum by:

$$\Delta\omega_{B_{12}}^{\text{UFZ}} = \Delta\omega_{B_{12}} \frac{G_{\text{acq}}}{G_{\text{sat}}} \tag{14}$$

This shows that for a fixed acquisition gradient the biggest apparent separation between the dips is obtained by minimizing the saturation gradient. The price to pay is that the dips or one of the dips can exit from the magnetization envelope if the condition $2|\omega_i - \omega_{B_{1,2}}|/G_{\text{sat}} > \gamma.r$ is encountered.

## 5 Areas for improvement

The use of Hyper-CEST experiments at low magnetic field (1 T) for detection of low quantities of $^{129}$Xe NMR-based biosensors is possible, even with a spectrometer not dedicated to $^{129}$Xe observation. Such a benchtop spectrometer, placed close to the optical pumping setup and near the imager could represent a helpful tool to i) know the xenon hyperpolarization level and ii) to know the thermodynamics and kinetics of the complex between the noble gas and the biosensor.

However, some issues have been encountered with these experiments. The most important is the difficulty to keep the hyperpolarization between two scans. Indeed shaking the NMR tube in the absence of a magnetic field high enough and sufficiently homogenous leads to fast depolarization. For each introduction of laser-polarized xenon into the NMR tube, we were limited to two to three experiments. This fast relaxation being linked to the diffusion of the noble gas into field gradients (in amplitude and in direction, see Cates et al. (1988)), a solution to counter this would be to increase the pressure inside the NMR tube, by increasing the amount of hyperpolarized xenon or by adding another gas such as nitrogen. Another solution



under study would be to build an homogenous field tunnel between the production site of the hyperpolarized species and the
spectrometer, such as in dissolution Dynamic Nuclear Polarization (Milani et al., 2015). In any case, the ultrafast version of
this experiment, the UFZ-spectroscopy, exhibits several advantages with respect to the classical version of the Hyper-CEST.

Working with a spectrometer not tuned to the frequency of interest was only possible as i) it is a low field, ii) the lowering of
the magnet temperature could improve the situation. However some concern may still appear when using a long rf saturation in
the $^{129}$Xe UFZ experiments, and every solution to bring the radio frequency of the $X$ channel of the spectrometer closer to the
resonance frequency of the nuclei of interest is appealing. An easy and low-cost solution was found through placing a solenoid
around the NMR tube, or even better by wrapping two copper foils around the NMR tube on either side of the region to be
detected, as displayed in Fig. 10a-b, a setup inspired by Wheeler and Conradi (2012). The resulting inductive tuning decreases
the inductance, and thereby induces a raise on the resonance frequency of the circuit.

Figure 10c-d shows the wobble curves obtained with the naked NMR tube (c) and with copper foils manually adjusted on
the NMR tube axis (d). Remarkably, the quality factor of the probe is not degraded by the presence of the copper foils, and
even a better matching is obtained. At the $^{129}$Xe frequency, the radiofrequency reflected amplitude at the probe which is ca. 35
$\mu$V without the foils (Fig. 10c) falls to 3 $\mu$V with them (Fig. 10d).

As can be seen on Fig. 10b, given the gap between the two copper foils, which should be smaller than the original detection
zone, a partial rf shielding of the sample must occur. This decreases the detection volume accordingly.

Note also that such a simple setup gives access to all nuclei in the region 10.9 - 12.1 MHz, *i.e.* $^{13}$C, $^{27}$Al, $^{23}$Na and $^{129}$Xe. It
is *a priori* useable on every benchtop spectrometer, as they are commonly based on a Halbach magnet for the static magnetic
field and a solenoid for the rf coil, which principal axis is colinear to the NMR tube and the magnet aperture. Creating a
inductive coupling with this arrangement is therefore an easy task.

*Author contributions.* PB conceived and designed the experiments. KC had the idea and developed the coils to shift the working frequency of
the benchtop spectrometer. EL, PB and CB performed the NMR experiments at high and low magnetic field. HD and PB wrote the simulation
programs. PB and EL processed the data. PB wrote the manuscript, which was reviewed by all the authors.

*Competing interests.* The authors declare no competing interests.

*Acknowledgements.* The authors warmly thank Thierry Brotin (ENS Lyon, France) for providing samples of cryptophanes, Jean-Claude
Berthet (CEA Saclay, France) for the preparation of the SEOP cells, and Craig Eccles (Magritek, Germany) for his help on the SpinSolve
Expert software. Support from French National Research Agency (ANR) is greatly acknowledged (Project-ANR-17-LCV2-0002 DESIR,
and Project-ANR-19-CE19-0024 PHOENIX).



**Figure 10.** Example of effect observed on the wobble curve of the X channel on the Magritek Spinsolve Carbon 43 when simple copper foils are wrapped around the NMR tube. a): Scheme of the setup, with the copper foils wrapped around the NMR tube ; b) Picture of one of these tubes. As we were limited to an outer diameter of 5 mm, the lower part of the NMR tube receiving the foils was only 4 mm ; c) Wobble curve of the $X$ channel without the copper foils ; d) Wobble curve with the copper coils.



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
