# Peer review of "129Xe Ultrafast Z-spectroscopy enables micromolar detection of biosensors on a 1T benchtop spectrometer"

_Magnetic Resonance, 2021_

## Author Response (AR1)

Dear Editor,

Please find our manuscript revised according to the comments posted on the website.

You will find below the comments of V.V. Telkki in bold italics, and our answers.
* * *
*The manuscript describes the first application of 129Xe Hyper-CEST method with a low field NMR spectrometer. The authors prove the feasibility of the experiments by comparing ultrafast Z-spectra of a solution including a mixture of two different cryptophane cages measured at 11.7 and 1 T. They show that bound Xe signal is observable even at 19 um concentration at 1 T. The manuscript describes lots of useful practical information related to the low-field experiments; for example, they tuned a Magritek Spinsolve Carbon spectrometer to Xe frequency by adjusting the temperature of the magnet, and they demonstrate also an alternative way of tuning by wrapping a copper foil around the sample. Furthermore, they compare experimental observations with the spectral simulations. The manuscript is very well written.*

*I recommend a minor revision based on the comments below.*

*Fig. 2 is not referred in the manuscript text, please refer.*

Well observed. A sentence will be added.

*Page 4: Sentence "This was expected as the magnet, of the Halbach type, delivers an horizontal static magnetic field, and thus xenon crosses areas of null field during its transfer" is bit illogical, because the horizontal direction of the magnetic field does not automatically mean that field is zero during the transfer of the sample. Do you mean that the fringe field on top of the instrument is close to zero?*

Yes this is partly what we mean. Whereas on the 11.7 T (non-shielded magnet) the magnetic field lines largely exceed the top of the magnet by which the sample is introduced and the sample pathway follows the magnetic field lines, here on the 1 T magnet which field lines are horizontal in the center, the magnetic field decreases much more rapidly and xenon crosses areas of null field during its transfer.

*Section 2.3: It would be interesting to see a comparison of the SNR values of the dissolved Xe peak in the spectra measured at 11.7 and 1 T (SNRs determined from the spectra shown in Fig. 3) to have an idea about the order of magnitude, how much lower the sensitivity at 1 T is. I understand that this value is dependent on many instrumental and sample transfer related factors, but still I would be curious to see the values.*

As both spectra of Fig. 3 have been processed in Bruker Topspin software, the SNRs can effectively be given. However, the reader will have to take precaution with this comparison for the reasons above mentioned; it effectively does not simply scale as expected (square root of the magnetic field).

*Fig. 5: as the authors discuss in the text, the profile is affected by the spatially dependent spin density, as the gradient is perpendicular to the sample tube, and the coil excitation-detection profile. The authors might discuss about the possibility for the spin-density and coil excitation-detection profile correction as described, e.g., in Ahola, Nat. Commun. 2017, 6, 8363.*

It is sure that for quantitative analysis of the CEST effect in this ultrafast experiment both the spin-density and the excitation-detection profile of the coil have to be considered; to implement a correction similar to what was proposed in this article would perhaps involve using (off-on)/off in the processing of each frequency coordinate. Although this is slightly beyond the scope of this paper, we will add a sentence about this point.

Also, we are not the first authors to use spatial encoding and low-field NMR. We have forgotten to cite the complete work of Gouilleux et al. in this domain, which is summarized in J. Magn. Reson. 319 (2020) 106810. We sincerely apologize to them; a sentence and this reference will naturally be added in the text.

→ *Thank you very much for the good and comprehensive replies to my comments. Related to your third last reply, spatial encoding has been used with low-field NMR instruments (and DNP hyperpolarization) also in the following works: King, Angew. Chem. Int. Ed. 2016, 55, 5040 and King, Chem. Sci. 2018, 9, 6143.*

→→ Thank you for bringing these articles to our attention, the references will be added in the revised manuscript.

*The shapes of the simulated spectral lines in Fig. 8 are quite difficult to see due to overlapping lines, could the plot be modified?*

It is difficult to find an ideal angle of view; this one seemed to be the best. And making the surface opaque would hide a lot of things. The only solution we propose is to put a color gradient according to the intensity and to add a contour plot.

*The authors might consider adding the script of the ultrafast Z-sequence to the supporting information of the manuscript.*

This is a good idea. This script will be added as supporting information in the revised version.
* * *
All the announced changes have been made on this revised manuscript. We will be ready to send the original Figures as well as the TeX source file when it will be requested.

Yours sincerely,

Patrick Berthault

---

## Author Response (AR2)

Dear Fabien,

Please find the last version of our manuscript modified according to your suggestions. Our institute, the CEA, is not very advanced in developing a system for data deposit, so we cannot satisfy your proposal for the moment.

Best regards,

Patrick